# Apply DEMATEL to Analyzing Key Barriers to Implementing the Circular Economy: An Application for the Textile Sector

**Wen-Kuo Chen** [1], **Venkateswarlu Nalluri** [2], **Hsing-Chun Hung** [2], **Ming-Cheng Chang** [2] and **Ching-Torng Lin** [3,*]

1    Department of Marketing and Logistic Management, College of Management,
     Chaoyang University of Technology, Taichung 413310, Taiwan; wkchen@cyut.edu.tw
2    College of Management, Da-Yeh University, Changhua 51591, Taiwan; nallurivenkey7@gmail.com (V.N.);
     hsc050219@gmail.com (H.-C.H.); lifestve@hotmail.com (M.-C.C.)
3    Department of Information Management, Da-Yeh University, Changhua 51591, Taiwan
*    Correspondence: charllin@mail.dyu.edu.tw

**Abstract:** Continuous improvement and innovation are solid foundations for the textile sector to maintain excellent growth and active sustainability. As the limited resources possessed by textile companies generally result in the incapability of implementing circular economy (CE) strategies simultaneously, recently, researchers advocate that organizations should analyze the influential inter-relationship between key barriers to explore the more dominant determinants for designing improved actions for implementing CE in the textile sector. CE implementation in the textile sector appears to be in its infancy. Although much attention has been paid to CE implementation barriers, the present study tries to fill this research gap by analyzing the causal relationships among the CE barriers in the textile sector. Therefore, the twelve barriers are identified by an extensive literature review, and the application of the Fuzzy Delphi Method (FDM) based on the expert options from the textile sector. Subsequently, the causal inter-relationship among the key CE barriers is based on expert opinions using the decision-making trial and evaluation laboratory (DEMATEL). The results of this study indicate that three key barriers require quick action: "consumers lack knowledge and awareness about reused/recycle (B1)", "lack of successful business models and frameworks to implement CE (B3)", and "lack of an information exchange system between different stakeholders (B8)". In addition, the results provide significant managerial implications, including implementations of CE in the textile sector. Not only should the government build regulations and friendly laws and encourage environmentally-friendly materials but the textile companies should also focus or monitor the recycling methods and quality to overcome the CE implementation issues. In addition, this study contributes to the textile sector transition toward CE by using the novel methodology for determining and prioritizing the key barriers. Finally, this work would help top management and the practitioners to better design effective infrastructural strategies for the textile sector transition towards CE.

**Keywords:** circular economy (CE); key barriers; Fuzzy Delphi Method (FDM); decision-making trial and evaluation laboratory (DEMATEL); textile sector

## 1. Introduction

The circular economy (CE) has been receiving much attention in the popular discourse as well as in discussions by industry leaders, policymakers, and researchers. The implementation of CE is already underway, and it represents a promising solution to the issues of resource scarcity and waste disposal [1]. CE refers to the transition of business operations from the traditional linear take-make-dispose model [2] to a more sustainable system in which the creation of circular loops of waste flows, materials, and energy counteracts the damage caused by resource acquisition [3]. CE combines recycling, redesign, reduction, and reuse with present production and consumption activities, which require radical systemic changes in how products and materials are manufactured, used, and disposed of [4].

Moreover, because of the disruptive nature of the transition, the implementation of CE appears to be in its infancy [5]. The practice–theory gap is still unexplored because present approaches for delivering CE are neither clear nor certain [6].

In the present literature, just a few researchers have examined the implementation and strategies of CEs [7]. Principato et al. [8] found CE methods to reduce food loss and wastage of the pasta sector in developing countries, while Saavedra et al. [9] mentioned that CE practices, with an eco-industrial design process, have been used in developed countries. In further studies, Masi et al. [10] carried out a review of the literature to determine and discuss the opportunities and challenges of CE, Walzberg et al. [11] suggested an alternative model by conducting the study on the adoption of CE operations, Priyadarshini and Abhilash [12] established a link between theory and practice for a better assessment of CE operational principles, and Shen et al. [13] proposed CE theoretical models of waste management. Moreover, the modern business strategies' development, such as industry 4.0 [14], digital economy [15], and circular economy practices [16–18], have become increasingly popular in developed countries as a result of their positive impact on the growth of the economy and the environment. In developing countries, the issues outlined are common in the textile supply chain across domains, including design, source procurement, fiber and clothing production, packing and delivery, usage and restoration, and waste management [4]. Shen et al. [13] believed that the input and output of the fashion sector's "textile product life cycle" had an influence on the environment, but the scale of the effect was astonishing. Saavedra et al. [9] stated that part of the reason is the huge scale of the textile sector, which is believed to be a $1.5 trillion sector, and the third largest manufacturing sector in the world, following automobiles and technology (House of Commons Environmental Audit Committee, 2020). Moreover, Cristea et al. [19] confirmed that the greenhouse gas emissions from textile production exceed the combined emissions from international aviation and maritime transport. If the emission of the textile sector continues along this path, it is expected that it will account for a quarter of the world's carbon emissions by 2050 [17]. In addition, the textile sector is facing tremendous resource and environmental issues due to the extreme pollution and waste generated from the modern or fashion phenomenon. The textile sector has major environmental effects, including the use of considerable quantities of chemicals and water, substantial generation of waste, and high use of greenhouse gases, which have improved the interest in increasing recycling and reuse practices in the textile sector [20]. Moreover, differentiating issues between contexts is another main research opportunity that can aid the implementation of CE and formulation of the relevant policy. In current CE literatures, they mostly focused on establishing a general framework of CE barriers [21] while some studies have mentioned the need for exploring barriers in specific business models or sectors [22–25]. Yet, the supply chain of the textile sector is a significant contributor to the global economy, and its system of distributing raw materials, producing, and consuming clothing operates majorly in a traditional or linear manner [26].

A circular economy is considered a stepping-stone to improve the production and consumption system for the textile sector [27,28]. By pushing the textile industry toward CE, it can potentially reduce the production of raw materials and reengineering processes in the product lifecycle in the textile sector and thus decrease the environmental effects in the textile sector [29]. In addition, the implementation of CE in the textile sector is still facing a few issues, and a systematic analysis of the different barriers related to the industry has yet to be undertaken. However, there is very little in the literature on the implementation of CE in the textile sector. Therefore, the present study has proposed a novel approach for analyzing CE implementation barriers. As for some similar studies, Schroeder et al. [30] analyzed CE practices for the agriculture sector, and Liakos et al. [31] studied manufacturing companies, whereas Tunn et al. [32] have introduced a new business model for sustainable consumption with CE. Bullock et al. [33] analyzed the issue of setting policies for CE implementation. Rossi et al. [34] built a performance measure of CE outcomes for plastic sector applications, and a review of the literature for success and failure factors, drivers, and practices related to CE [35,36]. As had happened among

previous studies of challenges, drivers and barriers are anticipated to be present in case of the implementation of CE [37]. Based on the above conditions, it is necessary to determine these key barriers of CE implementation in the textile sector. It is important to comprehend the primary barriers to CE adoption in the textile supply chain [38]. This supports the current study to determine and analyze the key barriers to the implementation of CE. Determination and analysis of the cause–effect relationships among barriers in the textile sector would support policymakers, manufacturers, and other stakeholders. Therefore, the present study's main objectives are the following:

to determine barriers to CE adoption in the textile sector by evaluating the current literature and incorporating expert opinions to select the appropriate barriers;

and, using the DEMATEL model, develop cause-and-effect relationships between key CE implementation barriers.

In order to achieve the present study objectives, the current study begins with the review of the literature to identify the barriers related to the implementation of CE. To further conform to the barriers, a set of questionnaires was prepared and expert options were taken of textile sector experts. After that, we performed the Fuzzy Delphi Method (FDM) to find key barriers with help of the experts' assessments from different textile companies in Taiwan [39]. Finally, the DEMATEL method is used to analyze the cause–effect relationships between key barriers [40]. Based on DEMATEL findings, barriers to CE adoption have been categorized into cause and effect groups. In fact, the DEMATEL approach was first developed in 1976 by the Battelle Memorial Institute of Geneva's research for the science and human relations program [41]. It is a powerful causal analysis technique that allows researchers to classify any criteria of the system into cause and effect groups [42]. Moreover, the DEMATEL approach can analyze the inter-relationships between the barriers. In addition, this technique helps to develop a graph depicting the cause–effect interaction within barriers or criteria. This can be used to identify and resolve complex issues [43].

According to the DEMATEL analysis, the results show that "consumers lack knowledge and awareness about reused/recycle (B1)", "lack of successful business models and frameworks to implement CE (B3)", and "lack of an information exchange system between different stakeholders (B8)" are the most affecting barriers to the implementation of CE in the textile sector. In addition, "lack of support supply and demand network (B4)", "high short- term costs and low short-term economic benefits (B11)", and "make the right decision to implement CE in the most efficient way (B12)" are minor barriers affecting the implementation of CE in the textile sector. The contribution of the present study is twofold. First, to the best of our knowledge, it is the first step in identifying a list of key barriers that must be overcome in order to implement CE practices in the textile sector. Secondly, the ongoing study uses an integrated technique of FDM and DEMATEL in order to have a better understanding of the relative importance to managers and policymakers in the textile sector and cause–effect interrelationships among CE barriers. Thereby, the most influencing barriers would be provided by a novel approach and it will assist textile company executives and policymakers in developing effective strategies with limited resources. Furthermore, this study aims to examine grouping barriers by a cause–effect relationship graph; it will empower textile company executives and managers in implementing successful supply chain prevention strategies.

The rest of the paper is mapped out as follows. Section 2 includes a literature review to identify the barriers, followed by Section 3 (research methodology). The results and discussion are presented in Section 4. In Section 5, the conclusion, as well as final limitations, and future research directions are given.

## 2. Literature Review

CE has been conceptualized and defined differently by different researchers. The conceptual backbone of CE is the creation of a closed-loop system of materials, energy, and waste flows [44]. This reduces the consumption of virgin resources and the generation of

waste and pollution, which, in turn, results in resource recovery and efficiency [45]. In this study, we adopt the definition of CE from Scarpellini et al. [46], as follows. Looking beyond the present extractive industrial model with take-make-waste, a circular economy intends to redefine productivity, emphasizing positive society-wide benefits. It means eventually decoupling economic activity from the consumption of the finite resources and reducing waste from the system. Underpinned by a transition to sustainable energy sources, the natural, social capital, and the circular model builds the economy. It is based on three strategic objectives: regenerating natural systems, design out waste and pollution, and keep products and materials in use. CE is an economic rather than environmental strategy. In the relevant literature, three directions have been adopted to explore CE implementation strategies: Section 2.1, CE implementation; Section 2.2, CE in the textile sector; and Section 2.3, identification of CE implementation barriers.

*2.1. CE Implementation*

From a systematic perspective, the implementation of CE has several practical aspects that vertically comprise the micro level, meso level, and macro level. These three levels are interdependent [47,48]. Research on barriers to the adoption of circular business models, e.g., Roos et al. [49], has focused on micro-level initiatives. For a focal firm, the challenge of moving toward a circular business model is selecting "the rationale of how a company creates, delivers, and captures value with and within closed material loops" [50]. Different business models, such as the product-as-a-service model, the resource recovery and circular supply models, and the product life extension model, all explore pathways toward CE [51]. Katz-Gerro and López Sintas [52] determined that there are seven types of barriers that discourage small and medium-sized businesses from implementing CE business models: lack of government support/effective legislation, lack of capital, administrative burdens, deficient corporate culture, lack of technical and technological expertise, lack of network support in supply and demand, and lack of information. Another survey-based analysis of 76 companies indicated that significant initial funding costs, as well as a misunderstanding and sense of urgency, are the main CE adoption barriers at the focal-firm level [53]. Moreover, Scipioni et al. [54] categorized barriers into those that are internal versus external to the focal firm, and they suggested modes of circular business models that require tailor-made CE solutions.

At the meso level, the creation of a closed-loop supply chain is crucial for various circular remanufacturing or recycling business models [55]. Ranta et al. [56] analyzed CE barriers in a supply chain and discovered that the most common barriers are "lack of consumer perception towards remanufactured products," "lack of public awareness of CE," and "technology limitation by the enterprises to make products that can be easily remanufactured." In addition, Bhatia et al. [57] analyzed the causal relationships among barriers in a closed-loop supply chain, and their results revealed that the most critical barriers exist in the remanufacturing and sales stages and that the elemental barriers are remanufacturers. Similarly, Rajput and Singh [58] recognized the important driving barriers as "interface designing" and "automated synergy model" in the supply chain context.

At the macro level, the main actors driving progress toward CE are legislative and governmental bodies. For instance, in China, the Cleaner Production Promotion Law of 2002 and the ensuing Circular Economy Promotion Law of 2009 formally introduced CE into the context of real-world public policy [59]. Moreover, the government of the Netherlands is considered the frontrunner in Europe in pursuing CE [60], and the European Commission has been implementing a variety of ambitious CE policies and launched the circular economy package in 2015, which was updated in the year 2018. In their large-N study on CE barriers in Europe, Kirchherr et al. [61] identified that a lack of consumer interest and awareness and a hesitant organization culture are especially significant CE barriers for business managers and policymakers. Modgil et al. [62] aggregated the findings in the literature and developed a CE framework according to "hard" and "soft" factors. Their results revealed that hard factors (e.g., those that are related to the availability of

technical solutions and financing) drive CE, whereas soft factors (e.g., social, regulatory, or institutional) inhibit the implementation of CE.

### 2.2. CE in the Textile Sector

Early research on CE barriers under different circumstances provided insight and practical guidance for CE implementation. Studies that are more recent have focused on specific contexts, such as the manufacturing sector [63], the automobile industry in emerging economies [64], and the construction and demolition waste management industry [65]. Except for these studies, studies on which barriers impede the textile sector from transforming toward CE and how they do so have been rare.

Textile consumption, along with those of food, housing, and transportation, has substantial environmental effects [66]. From a sustainability perspective, this sector's challenges include the reduction of its material and energy intensity, reduction of toxic substance dispersion, enhancement of recycling, maximization of renewable resource use, extension of product durability, and improvement of service intensity [34]. From a circular perspective, the status quo of the common linear flow of materials requires reform. Such reforms should be undertaken through the careful design of products and industrial processes in such a manner that materials are akin to perpetually flowing nutrients that are managed in closed loops [56]. Most recently, issues related to textile reuse and recycling have gained increased attention in the literature [67,68]. Textile reuse and recycling are generally recognized as superior methods for reducing environmental effects compared with incineration and landfilling [69]. However, textile recycling remains limited due to the sector's many socioeconomic challenges. The CE is the method of converting supply chain (SC) operations from a linear to a circular production/business model, through which used/waste materials and components are reintroduced by the SC through a closed-loop system by reusing, recycling, remanufacturing, reconstruction, and refurbishing as a way of recapturing inventory and reducing negative impacts [70,71]. With the adoption of CE strategies, waste generation from manufacturing can be reduced by a significant amount, (1) as well as CE implementation barriers to the creation of innovative sustainable business models. (2) In a CE, waste materials are assessed for further use in the fashion industry, including the inability to rethink the design phase for sustainable product development, poor consumer education, low consumer expectations regarding sustainability, and a lack of alignment of values along the supply chain.

In general, transforming the textile sector toward a new CE requires system-level changes with an unprecedented degree of commitment, collaboration, and innovation [72]. Additionally, the speed and scale of the transition depend on the knowledge, awareness, and engagement of all market participants. Although various barriers have been revealed by these prior studies, scholars have provided few suggestions for transforming textile systems [73]. Our study thus aims to contribute to this gap by providing prescriptions for the development of CE intervention strategies for the textile industry.

### 2.3. Identification of CE Implementation Barriers

As shown in Table 1, the barriers to the implementation of CE in the textile sector were identified by conducting a systematic review of the literature and a textile company expert's opinion. For example, Leal Filho et al. [74] identified six barriers that impede textile recycling: (1) economic viability, (2) composition of textile products, (3) poor coordination or weak policies, (4) technological limitations, (5) limited public participation and lack of information, and (6) lack of recyclable textile materials or inadequate standards. Similarly, Larney et al. [75] proposed five barriers to textile recycling: (1) lack of motivation and propagation, (2) lack of policies and regulations, (3) lack of awareness, low cost, and fashion press, (4) availability of collection bins and containers, and (5) a wide variety of materials and chemicals used in the production of textiles. These barriers constitute the massive challenges that the textile sector faces at the macro and meso levels. However, barriers also exist at the micro-level. Wu et al. [76] identified several entrepreneurial challenges that

are potential barriers to a paradigm shift toward CE. Then, the FDM was used after the literature review to identify twelve barriers, which also included organized interactions between expert groups on the proposed key barriers. It took three rounds of changes and revisions before arriving at the shortlisted key barriers, which are enlisted in Table 1. The team of experts consisted of spinning mills, fabric mills, and finishing of textiles, textile products manufacturing, and wearing apparel and clothing accessories manufacturing as shown in Table 2.

**Table 1.** The CE barriers with reference.

| S.no | Barriers Name | Brief Description | Reference |
|------|---------------|-------------------|-----------|
| B1 | Consumers' lack of knowledge and awareness about reused/recycle | This barrier refers to customer attitudes and knowledge toward fashion on recycling methods. | [77–81] |
| B2 | High purchasing cost of environmentally friendly materials | This barrier refers to the general public that would endorse, be obligated, and be engaged in purchasing eco-friendly clothing. | [77,78,81] |
| B3 | Lack of successful business models and frameworks to implement CE | This barrier refers to the absence of guidelines and models on refurbishment and recycling performance assessment. | [77,79,81] |
| B4 | Lack of support supply and demand network | This barrier refers to measures of the complexity throughout the supply chain (specifically in its logistical, financial, and legal aspects), which, in turn, affect the value chain of a product, process, or service. Therefore, the need to close the traditional supply chain loop can cause significant dynamic complexity and deep uncertainty. | [79,81,82] |
| B5 | Obstructing laws and regulations | This barrier indicates the obstructive laws and regulations and unsupportive laws on waste management from government authorities. | [77–82] |
| B6 | Design challenge to reuse and recovery products | This barrier refers to the firms facing problems related to the quality of products in circulation containing recycled materials or products being refurbished. | [77–81] |
| B7 | Limited availability and quality of recycling material | This barrier includes technological limitations to tracking recycled materials as well as in maintaining the quality of products made from recovered materials, in designing reused and recovered products, and ensuring a safe return to the biosphere. | [77,78,80–82] |
| B8 | Lack of an information exchange system between different stakeholders | This barrier refers to the role of information in the implementation of CE at an optimal efficiency, and the lack of an information exchange system between different stakeholders. | [77,80–82] |
| B9 | Unclear vision in regards of CE | This barrier indicates a lack of standardization, recycling policies in waste management that fail to result in high-quality recycling, and results in an unclear vision regarding CE. | [79] |
| B10 | Insufficient internalization of external costs | This barrier is identified as the limited funding for circular business models, insufficient internalization of external costs, difficulties in establishing correct product prices, high upfront investment costs, high short-term costs but low short-term economic benefits, limited availability and quality of recycled materials, high cost of environmentally friendly materials, and increasing production costs. | [77] |
| B11 | High short-term costs and low short-term economic benefits | This barrier refers to the affordability of circular products which is undermined when the price of virgin materials is much less than that of environmentally friendly materials and when the costs of manufacturing circular products are increasing. Textile recycling is limited to low-value applications because of the substantial variation in the composition of different types of fibers, dyestuffs, and chemicals used in finishing. | [77,81,82] |
| B12 | Make the right decision to implement CE in the most efficient way | This barrier indicates that decisions requiring new sustainable production and close partnerships are essential in the development process of technical solutions considering the requirement to communicate with industry stakeholders regarding these strategies. | [77,78,82] |

**Table 2.** Profile of experts.

|  |  | Sample Size |
| --- | --- | --- |
| Industry category | Fabric Mills | 3 |
|  | Yarn Spinning Mills | 3 |
|  | Finishing of Textiles | 3 |
|  | Non-woven Fabrics Mills | 4 |
|  | Textile products Manufacturing | 5 |
|  | Wearing Apparel and Clothing Accessories Manufacturing | 4 |
| Employees of the firm | <100 | 4 |
|  | 100–300 | 4 |
|  | 300–500 | 1 |
|  | 500–1000 | 6 |
|  | >1000 | 7 |
| Work experience in textile sector (years) | <10 | 6 |
|  | 11–15 | 0 |
|  | 16–20 | 6 |
|  | >20 | 10 |
| Work experience in current employed company (years) | <10 | 6 |
|  | 11–15 | 1 |
|  | 16–20 | 7 |
|  | >20 | 8 |

## 3. Methodology

This research purpose is to establish the cause-and-effect relationships among the CE key barriers of the textile sector. The methodology of this study is shown in Figure 1. To achieve the objectives of the present study, we used the DEMATEL approach, which incorporates the textile sector experts in a well-defined and structured manner to determine the cause–effect relationship between the barriers. DEMATEL leads to be significant in promoting the internal validity of the results [83]. This approach has been used effectively by few researchers, such as Kumar and Mathiyazhagan [84] for implementing lean manufacturing, Sharma et al. [85] for IT enablers for the manufacturing sector in India, and Li et al. [86] for modeling drivers of the textile-selecting suppliers. For example, Chen et al. [87] used a combination of ISM and MICMAC analyses and mentioned the drawbacks of the ISM. In addition, few studies suggested, such as Kaur et al. [88], applications of the DEMATEL analysis. Moreover, the present research flowchart mainly depicts the three-stage process structure as shown in Figure 1.

**Stage 1:**

In stage 1, as a part of the exacting research methodology (left part research flowchart as shown in Figure 1), the barriers are continuing to hinder the textile companies from effective implementation of CE. According to a few research studies [89], there are still some barriers to stay alive both inside and outside of industrialized organizations. Completed typical barriers can be found in the literature to further understand the CE implementation barriers. [28]. In Section 2.3, we examined a wide range of research articles on CE implementation issues and their associated barriers. In order to recognize and investigate the issues and barriers related to CE implementation, the authors conducted a systematic review of

the existing literature by searching Google Scholar, springer databases, Scopus, and science direct. The following keywords were used for the search, including circular economy (CE), CE implementation, barriers for CE implementation, and the textile sector. The literature review adopts those existing between 2001 and 2021, as the review of literature related to CE implementation is covered [6]. During the survey, 68 research articles from journals related to implementation were shortlisted. Based on these articles, the 12 significant barriers to CE adoption were identified in the textile sector, as shown in Table 1. This was followed by confirming the identified barriers from different textile companies' experts in Taiwan. The profiles of the experts are shown in Table 2. Then, we collected options from the textile experts by issuing questionnaires. Participants were asked to rate each item on a five-point scale, i.e., strongly agree, agree flair, disagree, and strongly disagree for 1, 2, 3, 4, and 5, respectively. We received 80% valid responses among all. As shown in Table 2, the experts are from different textile companies. Most of the experts have less than twenty years of experience in their present working companies and all have more than twenty years of working experience in the textile sector. Next, the FDM was performed to find the key barriers based on the experts' assessments, from textile companies in Taiwan [42]. Twelve key barriers were identified based on the threshold value (0.60) of the FDM results as shown in Table 3. At the present stage, the final step, we commented on the textile companies' experts with a set of survey questions to examine the cause–effect relationship among key barriers.

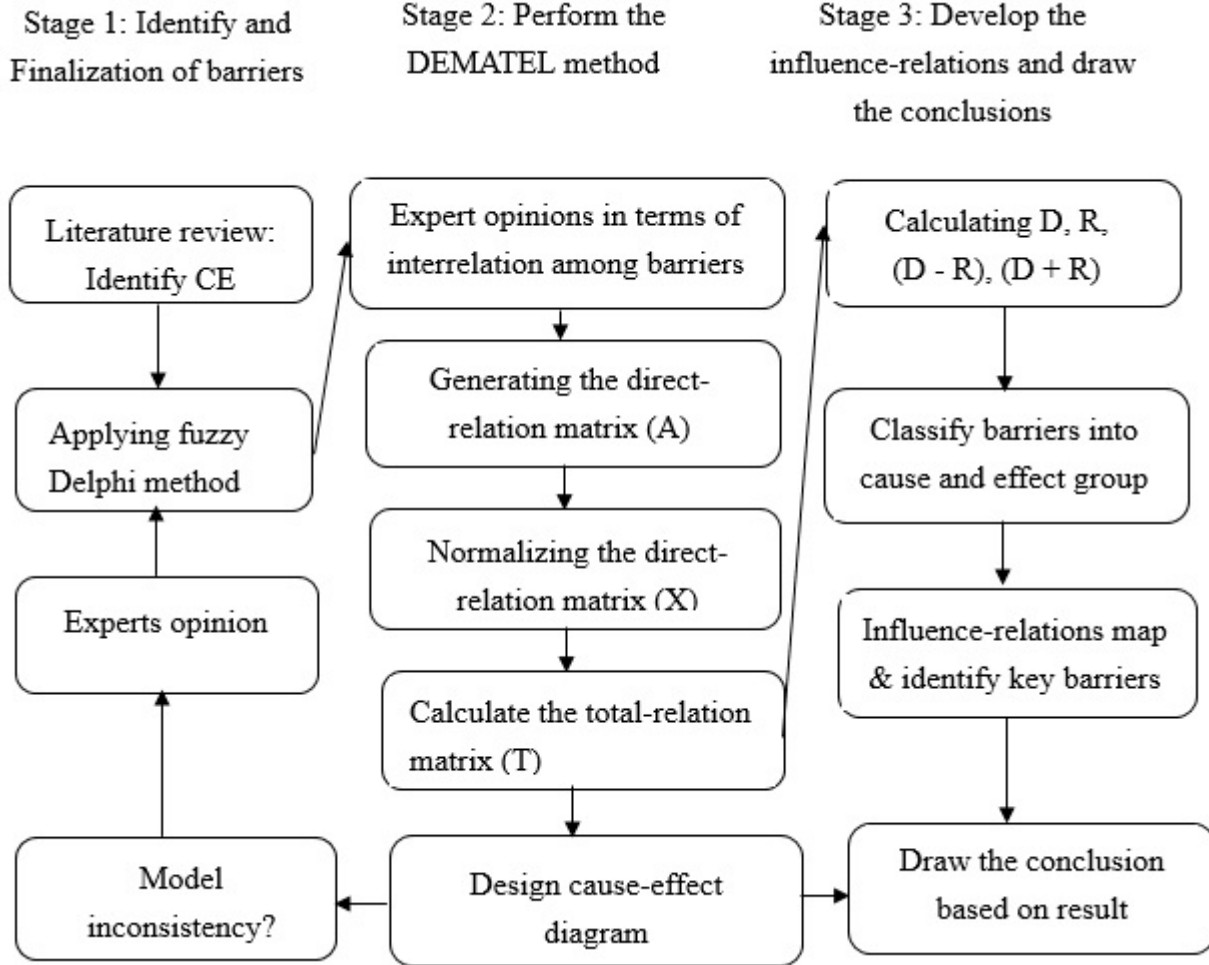

**Figure 1.** Research flowchart.

**Table 3.** The selected key barriers based on Fuzzy Delphi Method result.

| S.no | Barriers Name | FDM Threshold Value at 0.60 |
|------|---------------|------------------------------|
| B1 | Consumers' lack of knowledge and awareness about reused/recycle | 0.64 |
| B2 | High purchasing cost of environmentally friendly materials | 0.62 |
| B3 | Lack of successful business models and frameworks to implement CE | 0.61 |
| B4 | Lack of support supply and demand network | 0.62 |
| B5 | Obstructing laws and regulations | 0.63 |
| B6 | Design challenge to reuse and recovery products | 0.63 |
| B7 | Limited availability and quality of recycling material | 0.64 |
| B8 | Lack of an information exchange system between different stakeholders | 0.61 |
| B9 | Unclear vision in regards of CE | 0.60 |
| B10 | Insufficient internalization of external costs | 0.61 |
| B11 | High short-term costs and low short-term economic benefits | 0.63 |
| B12 | Make the right decision to implement CE in the most efficient way | 0.61 |

**Stage 2:**

In addition, the DEMATEL method is utilized for analyzing and building a structural approach of a causal relationship between the identified barriers. The DEMATEL approach was established to resolve and learn the complex criteria and intertwined issues group in 1976. DEMATEL is a well-known methodology that is often used for the assessment of decision problems in Japan [86]. The DEMATEL illustrates the causal relationships among the causes and effects of various variables and provides a structural framework for the system. DEMATEL has a significant advantage over other models in that it enables producing potential findings with the minimum information [84]. Although few other approaches can also be used for the analyzing of factors, such as interpretive structural modeling (ISM) and analytical network process (ANP). In comparison to ISM, the DEMA-TEL method assists in the identification of contextual relationships between factors and emphasizes the impact of their interactive relations. Further, this approach also determines the proportion of the cause-and-effect relationships of the barriers [90]. The DEMATEL is useful for not only describing direct subsystem relationships but also for defining the degree to which the subsystems interact. Moreover, if we want to measure or evaluate the cause–effect interaction between subsystems, DEMATEL seems to be more useful than the ISM in the measurement of complex systems [87]. The DEMATEL approach not only transforms interdependency interactions into a cause-and-effect cluster via matrixes but also discovers the essential barriers of an intricate system of barriers with the aid of an impact association graph [91]. Researchers may use the DEMATEL approach to understand the conceptual interactions between the barriers used within the issue structure and to assess the determination of their cause–effect relationships, compared with other modeling approaches like total interpretive structural modeling (TISM), graph theory and matrix approach (GTMA), and ANP [92].

**Step1: Generating the Direct-Relation Matrix (A)**

After preparing the list of relevant barriers or criteria, subject to the DEMATEL scale, every expert was asked to make pairwise comparisons between one barrier with another barrier. After that, any individual options and assessments about the causality among one barrier with another barrier were obtained from each expert's initial-relation matrix by using Equation (1). The scale ranging from 0 to 4 was used, which indicates no influence, very low influence, low influence, high influence, and very high influence to illustrate the inter-relationship among the identified barriers as shown in Table 4. The same method

would be following to fill out all of the experts' options as shown in Equation (1). Indeed, there are p experts where p = {1, 2, 3 ... *n*}. The equation [93] is as follows:

$$
A_p = \begin{bmatrix}
0 & a12 & a13 & \dots & .a1(n-1) & a1n \\
a21 & 0 & a23 & \dots & a2(n-1) & a2n \\
\dots & \dots & \dots & \dots & \dots & \dots \\
\dots & \dots & \dots & \dots & \dots & \dots \\
a(n-1)1 & a(n-1)2 & a(n-1)3 & \dots & 0 & a(n-1)n \\
an1 & an2 & an3 & \dots & an(n-1) & 0
\end{bmatrix} \tag{1}
$$

where, Ap determines each expert interaction option among barriers.

**Table 4.** The correspondence of the DEMATEL scale.

| Linguistic Terms | Numerical Value |
|:---:|:---:|
| No influence | 0 |
| Very low influence | 1 |
| Low influence | 2 |
| High influence | 3 |
| Very high influence | 4 |

**Step2: Normalizing the Direct-Relation Matrix (X)**

In this step, the normalizing direct-relation matrix (X) is computed in this process. It is possible by using the formula given in the following equation:

$$
X = k \cdot A \tag{2}
$$

$$
\text{Here } k = \cfrac{1}{\underset{1 \le i \le n}{max} \ \sum_{j=1}^{n} a_{ij}}, \ i, j = 1, 2, \dots, n \tag{3}
$$

where A indicates the initial-relation matrix as per Equation (1), k is the average of $a_{ij}$ of all experts, and X denotes the normalized direct-relation matrix. It should be observed that for the DEMATEL approach to be feasible, the number of each column in the normalized direct-relation matrix must be less than one [94].

**Step3: Calculate the Total-Relation Matrix (T)**

In this step, the total-relation matrix (T) is evaluated using the following equation:

$$
T = X (I - X)^{-1} \tag{4}
$$

where I denotes the identity matrix, T indicates the total-relation matrix, and X means the normalizing matrix as per the Equation (2).

The total-relation matrix T computes the sum of a number of rows (D) and the sum of a number of columns (R). D and R are calculated in the T matrix with the use of the following equations [40].

$$
(D) = [d_{ij}]_{n \times 1} = \left[ \sum_{j=1}^{n} d_{ij} \right]_{n \times 1} \tag{5}
$$

$$
(R) = [r_{ij}]_{1 \times n} = \left[ \sum_{i=1}^{n} r_{ij} \right]_{1 \times n} \tag{6}
$$

In addition, to obtain the threshold value ($\alpha$), all elements of the averages contained in the matrix T are added and divided by the number of elements present in the matrix. This computation is done by using the following equation:

$$\alpha = \frac{\sum_{j=1}^{n} \sum_{i=1}^{n} r_{ij}}{n^2} \qquad (7)$$

where the total number of elements in the total relation matrix T is represented by $n^2$. As the number of barriers = $n$, the number of total elements in matrix $T = n \times n = n^2$ [95].

Next, the linking diagram is created by plotting the values of (D + R) and (R − C). In this diagram, the $Y$-axis refers to the values of (D − R), while the $X$-axis refers to the values of (D + R). A driven graph is used to describe the interrelationships between the key barriers. The values in the T matrix that meet or exceed $\alpha$ are measured to have a high level of influence. The directed graph is created using the influential strength matrix.

## 4. Result and Discussion

In the current study, twelve key barriers of the textile sector to implementation of CE were identified based on a FDM threshold value of 0.60 as shown in Table 2. To understand the causal relationships between the key barriers and to determine the cause and effect barriers, a DEMATEL approach was adopted as shown in Figure 1.

As per the direct-relation matrix, (A) was developed using the experts' inputs and computed using Equation (1). The experts were given their options based on the linguistic terms scale. The range of the scale is 0 to 4, which is no influence, very low influence, low influence, high influence, and very high influence, as shown in Table 4. For example, there is a very high influence between the barrier B1 and B6, and value "4" has been placed in the cell (7, 1); meanwhile, there is no influence between the barriers B2 and B5, so the value "0" has been placed in the cell (6, 3). The result of the direct-relation matrix (A) of the pair-wise comparison of barriers' influences are captured in Table 5.

**Table 5.** Generating the direct-relation matrix (A).

| Barriers | B1 | B2 | B3 | B4 | B5 | B6 | B7 | B8 | B9 | B10 | B11 | B12 |
|---|---|---|---|---|---|---|---|---|---|---|---|---|
| B1 | 0 | 3 | 3 | 3 | 2 | 4 | 4 | 3 | 2 | 4 | 4 | 2 |
| B2 | 3 | 0 | 3 | 3 | 0 | 4 | 4 | 2 | 2 | 3 | 4 | 3 |
| B3 | 4 | 3 | 0 | 3 | 1 | 3 | 4 | 3 | 3 | 0 | 4 | 3 |
| B4 | 2 | 3 | 3 | 0 | 2 | 3 | 4 | 2 | 1 | 3 | 4 | 2 |
| B5 | 3 | 2 | 2 | 2 | 0 | 3 | 2 | 0 | 1 | 0 | 2 | 3 |
| B6 | 4 | 3 | 3 | 3 | 0 | 0 | 3 | 2 | 4 | 2 | 4 | 2 |
| B7 | 2 | 4 | 3 | 3 | 3 | 4 | 0 | 2 | 1 | 3 | 4 | 2 |
| B8 | 4 | 2 | 3 | 3 | 2 | 3 | 3 | 0 | 3 | 3 | 3 | 3 |
| B9 | 4 | 2 | 3 | 2 | 2 | 3 | 3 | 3 | 0 | 4 | 3 | 2 |
| B10 | 2 | 4 | 3 | 3 | 0 | 3 | 3 | 2 | 2 | 0 | 4 | 2 |
| B11 | 2 | 4 | 3 | 2 | 0 | 3 | 2 | 2 | 2 | 3 | 0 | 4 |
| B12 | 2 | 4 | 2 | 3 | 0 | 2 | 2 | 2 | 2 | 2 | 3 | 0 |

Next, as per step 2 of the DEMATEL approach, the normalizing of the direct-relation matrix (X) has been calculated using Equation (2), where Equation (2) indicates the average options of the experts. Results of the direct-relationship matrix are shown in Table 6.

**Table 6.** Normalizing the direct-relation matrix (X).

| Barriers | B1 | B2 | B3 | B4 | B5 | B6 | B7 | B8 | B9 | B10 | B11 | B12 | $\sum_{j=1}^{n} aij$ |
|---|---|---|---|---|---|---|---|---|---|---|---|---|---|
| B1 | 0 | 0.08 | 0.08 | 0.08 | 0.05 | 0.11 | 0.11 | 0.08 | 0.05 | 0.11 | 0.11 | 0.05 | 34 |
| B2 | 0.08 | 0 | 0.08 | 0.08 | 0 | 0.11 | 0.11 | 0.05 | 0.05 | 0.08 | 0.11 | 0.08 | 31 |
| B3 | 0.11 | 0.08 | 0 | 0.08 | 0.02 | 0.08 | 0.11 | 0.08 | 0.08 | 0 | 0.11 | 0.08 | 31 |
| B4 | 0.05 | 0.08 | 0.08 | 0 | 0.05 | 0.08 | 0.11 | 0.05 | 0.02 | 0.08 | 0.11 | 0.05 | 29 |
| B5 | 0.08 | 0.05 | 0.05 | 0.05 | 0 | 0.08 | 0.05 | 0 | 0.02 | 0 | 0.05 | 0.08 | 20 |
| B6 | 0.11 | 0.08 | 0.08 | 0.08 | 0 | 0 | 0.08 | 0.05 | 0.11 | 0.05 | 0.11 | 0.05 | 30 |
| B7 | 0.05 | 0.11 | 0.08 | 0.08 | 0.08 | 0.11 | 0 | 0.05 | 0.02 | 0.08 | 0.11 | 0.05 | 31 |
| B8 | 0.11 | 0.05 | 0.08 | 0.08 | 0.05 | 0.08 | 0.08 | 0 | 0.08 | 0.08 | 0.08 | 0.08 | 32 |
| B9 | 0.11 | 0.05 | 0.08 | 0.05 | 0.05 | 0.08 | 0.08 | 0.08 | 0 | 0.11 | 0.08 | 0.05 | 31 |
| B10 | 0.05 | 0.11 | 0.08 | 0.08 | 0 | 0.08 | 0.08 | 0.05 | 0.05 | 0 | 0.11 | 0.05 | 28 |
| B11 | 0.05 | 0.11 | 0.08 | 0.05 | 0 | 0.08 | 0.05 | 0.05 | 0.05 | 0.08 | 0 | 0.11 | 27 |
| B12 | 0.05 | 0.11 | 0.05 | 0.08 | 0 | 0.05 | 0.05 | 0.05 | 0.05 | 0.05 | 0.08 | 0 | 24 |

Further, a total-relation matrix is calculated by using Equation (4) and is shown in Table 7. The T matrix is developed by rejecting the early significant relationship for attaining the noteworthy connection. So, the threshold value ($\alpha$) is formulated using Equation (7) to develop the causal diagram. Based on the threshold value, we can determine the significant and insignificant barriers [96]. The $\alpha$ value is computed as 0.53, and the barrier values in the T matrix less than $\alpha$ value (0.53) were neglected for further processing of the DEMATEL. In Table 7, barrier values equal to or more than the threshold value are shown as STARE mark.

**Table 7.** Calculate the total-relation matrix (T).

| Barriers | B1 | B2 | B3 | B4 | B5 | B6 | B7 | B8 | B9 | B10 | B11 | B12 | D |
|---|---|---|---|---|---|---|---|---|---|---|---|---|---|
| B1 | 0.57 * | 0.71 * | 0.65 * | 0.63 * | 0.27 | 0.74 * | 0.72 * | 0.52 | 0.49 | 0.62 * | 0.82 * | 0.57 * | 7.35 |
| B2 | 0.60 * | 0.59 * | 0.61 * | 0.59 * | 0.20 | 0.69 * | 0.68 * | 0.47 | 0.46 | 0.56 * | 0.77 * | 0.55 * | 6.82 |
| B3 | 0.63 * | 0.66 * | 0.53 * | 0.59 * | 0.23 | 0.67 * | 0.68 * | 0.49 | 0.48 | 0.48 | 0.77 * | 0.55 * | 6.82 |
| B4 | 0.54 * | 0.62 * | 0.57 * | 0.47 | 0.24 | 0.62 * | 0.63 * | 0.43 | 0.40 | 0.52 | 0.72 * | 0.49 | 6.30 |
| B5 | 0.42 | 0.43 | 0.39 | 0.38 | 0.13 | 0.46 | 0.42 | 0.26 | 0.29 | 0.31 | 0.48 | 0.39 | 4.41 |
| B6 | 0.62 * | 0.65 * | 0.60 * | 0.58 * | 0.20 | 0.58 * | 0.65 * | 0.46 | 0.50 | 0.53 * | 0.75 * | 0.52 | 6.68 |
| B7 | 0.57 * | 0.68 * | 0.59 * | 0.58 * | 0.27 | 0.68 * | 0.56 * | 0.45 | 0.42 | 0.54 * | 0.75 * | 0.52 | 6.65 |
| B8 | 0.64 * | 0.65 * | 0.62 * | 0.60 * | 0.26 | 0.68 * | 0.67 * | 0.42 | 0.49 | 0.57 * | 0.75 * | 0.56 * | 6.95 |
| B9 | 0.63 * | 0.64 * | 0.60 * | 0.56 * | 0.25 | 0.66 * | 0.65 * | 0.49 | 0.40 | 0.58 * | 0.74 * | 0.52 | 6.77 |
| B10 | 0.53 * | 0.64 * | 0.56 * | 0.54 * | 0.18 | 0.62 * | 0.61 * | 0.43 | 0.42 | 0.43 | 0.71 * | 0.49 | 6.22 |
| B11 | 0.51 | 0.62 * | 0.54 * | 0.50 | 0.17 | 0.59 * | 0.56 * | 0.41 | 0.41 | 0.50 | 0.58 * | 0.52 | 5.97 |
| B12 | 0.47 | 0.57 * | 0.47 | 0.48 | 0.16 | 0.52 | 0.51 | 0.38 | 0.37 | 0.43 | 0.60 * | 0.37 | 5.38 |
| R | 6.79 | 7.52 | 6.79 | 6.56 | 2.60 | 7.56 | 7.39 | 5.25 | 5.17 | 6.11 | 8.49 | 6.10 | |

Note: Threshold = * => 0.53.

**Stage 3: Producing a Causal Diagram**

In addition, D, R values were calculated by using Equations (5) and (6), as shown in Table 7. The results of D and R confirm the degree of relational influence among each key barrier, respectively. Then, the authors formulated (D + R) and (D − R) values as shown in Table 8. For example, calculations of (D + R) and (D − R) are for B1; the D-value is

7.35 and R-value is 6.79, so adding them together is 14.14 (D + R) and subtracting them is 0.56 (D − R).

**Table 8.** Prominence and relation results obtained by using the DEMATEL method.

| Barriers | D | R | D − R | D + R |
|---|---|---|---|---|
| B1 | 7.35 | 6.79 | 0.56 | 14.14 |
| B2 | 6.82 | 7.52 | −0.69 | 14.34 |
| B3 | 6.82 | 6.79 | 0.03 | 13.62 |
| B4 | 6.30 | 6.56 | −0.26 | 12.86 |
| B5 | 4.41 | 2.60 | 1.81 | 7.02 |
| B6 | 6.68 | 7.56 | −0.87 | 14.25 |
| B7 | 6.65 | 7.39 | −0.73 | 14.05 |
| B8 | 6.95 | 5.25 | 1.69 | 12.21 |
| B9 | 6.77 | 5.17 | 1.60 | 11.95 |
| B10 | 6.22 | 6.11 | 0.10 | 12.33 |
| B11 | 5.97 | 8.49 | −2.52 | 14.47 |
| B12 | 5.38 | 6.10 | −0.72 | 11.49 |
| Average | | | 0 | 12.73 |

As shown in Table 8, the barrier with a D–R value less than zero is identified as an effective group, while a barrier with more than the D–R value comes under the cause group. Based on the DEMATEL results as shown in Table 8 and Figure 2, the causal interactions and the degrees of influence among the CE adoption barriers in the textile sector are described as follows.

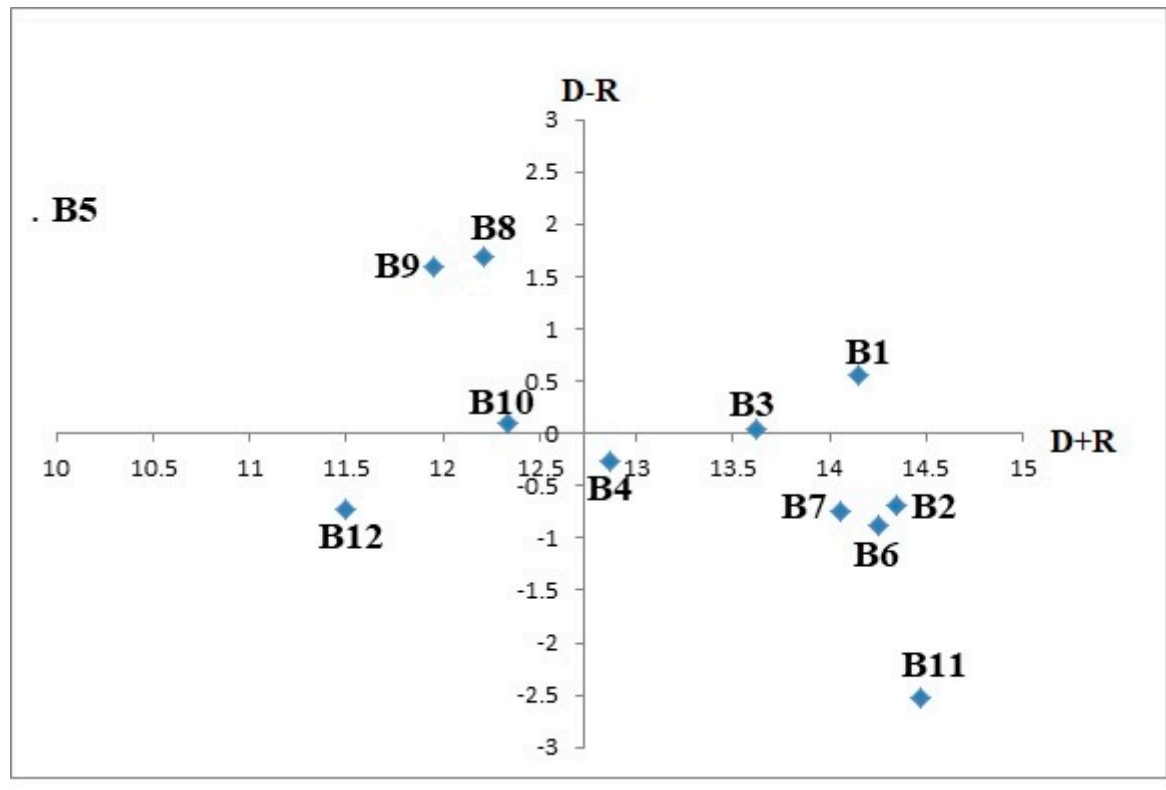

**Figure 2.** Cause-effect diagram.

Strong relation, high prominence: Consumers' lack of knowledge and awareness about reused/recycle (B1), and lack of successful business models and frameworks to implement CE (B3). These two key barriers were the case group barriers and were core factors that strongly influenced other barriers. Thus, they were the driving barriers for CE implementation in the textile sector.

Strong relation, low prominence: High purchasing cost of environmentally friendly materials (B2), lack of support supply and demand network (B4), design challenge to reuse and recovery products (B6), limited availability and quality of recycling material (B7), high short-term costs and low short-term economic benefits (B11). These five barriers slightly influence a few other barriers. It is indicated that these are relatively independent barriers.

Weak relation, low prominence: Make the right decision to implement CE in the most efficient way (B12). This barrier was slightly influenced by the other barriers, confirming that B12 is a relatively independent factor.

Weak relation, high prominence: Obstructive laws and regulations (B5), lack of an information exchange system between different stakeholders (B8), unclear vision in regards of CE (B9), and insufficient internalization of external costs (B10). These four barriers were the effect group barriers that were influenced by the remaining barriers. Despite requiring improvement, B5, B8, B9, and B10 could not be directly improved because they came under the effect group barriers.

According to the analysis results, six cause group barriers are identified, namely, "consumers lack knowledge and awareness about reused/recycle (B1)", "lack of successful business models and frameworks to implement CE (B3)", "obstructive laws and regulations (B5)", "lack of an information exchange system between different stakeholders (B8)", "unclear vision in regards of CE (B9)", and "insufficient internalization of external costs (B10)" are found to be the causal factors. Furthermore, "high purchasing cost of environmentally friendly materials (B2)", "lack of support supply and demand network (B4)", "design challenge to reuse and recovery products (B6)", "limited availability and quality of recycling material (B7)", "high short- term costs and low short-term economic benefits (B11)", and "make the right decision to implement CE in the most efficient way (B12)" are determined as the effect group. These are influenced by cause group barriers and affect the implementation of CE in the textile sector.

It may be noted that "consumers lack of knowledge and awareness about reused/recycle (B1)" has identified as a highly significant positive impact among all barriers, confirming that the lack of knowledge about the reusing materials among customers plays a key role in CE implementation of the textile sector. In addition, "lack of successful business models and frameworks to implement CE (B3)" is also a critical barrier among the 12 CE implementation barriers in the textile sector, and is the second criterion to consider; the following barriers are B2, B6, B7, and B11. These findings imply that these cause group barriers may improve the effect group barriers if the textile companies' top management can set up the information related to reuse/recycle measures to the supply chain members first and then design the reuse and recovery products. These steps would increase customer satisfaction and public reputation. Thus, the most effective group barriers or core issues including high purchasing cost of environmentally friendly materials (B2), design challenge to reuse and recovery products (B6), limited availability and quality of recycling material (B7) will be readily solved. The "insufficient internalization of external costs (B10)" is closer to the center among all barriers. It shows the identified causal group barriers and barriers that influence it less, namely, make the right decision to implement CE in the most efficient way (B12) was established to have a lower significance weight. As shown in Figure 3, the digraph of net cause and effect is drawn. Finally, after comparing with the threshold value as shown in Table 7, a directed graph for the barriers is created to show the relationship, as shown in Figure 3.

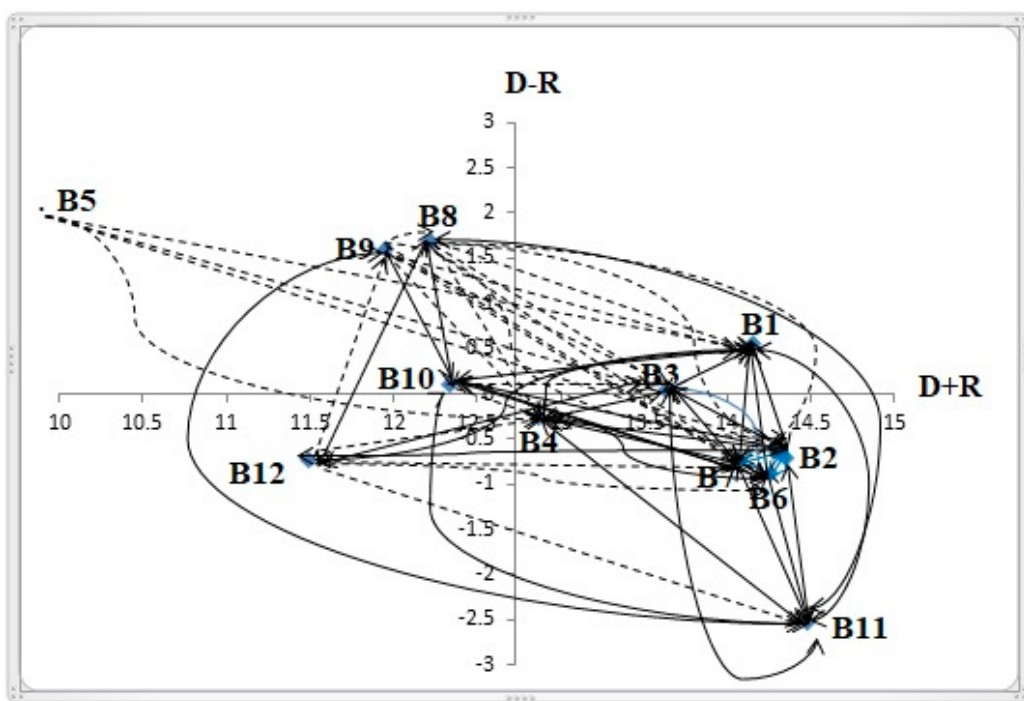

**Figure 3.** Causal inter-relationships of CE barriers.

Further, we draw the causal interrelationships graph of CE barriers in the textile sector based on the T matrix results, as shown in Table 7, and the interaction among the CE key barriers is shown in Figure 3. The double arrow-headed lines indicate the causal interactions among each pair of barriers, whereas single dotted-arrow lines represent less influence between each other, as shown in Figure 3. It has been found that "high purchasing cost of environmentally friendly materials (B2)" has strong interactions with another barrier, and the barriers for "design challenge to reuse and recovery products (B6)", "limited availability and quality of recycling material (B7)" are further related by highly influencing, or having more interactions with, other barriers of CE implementation in the textile sector. Therefore, the textile companies need to control those key barriers to implement CE in their textile supply chain.

## 5. Conclusions

The present study aims to determine the key barriers to the implementation of CE in the textile sector. To achieve the study objectives, the authors conducted an extensive review of the literature. In addition, the application of FDM was performed to find the key barriers. After that, we analyzed the interrelationships amongst key CE barriers with the DEMATEL approach. CE concepts have been identified as an important research topic in today's environmentally conscious world. The textile companies are unable to implement CE in their supply chain due to high global competition, demand for sustainable production processes, and increasing environmental problems. As per the recent past, scholars have correlated CE implementations in the textile sector [29]. For textile companies, some studies stated that in the textile industry, CE practices are used as an eco-industrial design process [97]. In addition, while Awan et al. [98] found that the CE refers to the transformation of business processes from the traditional linear "take-make-dispose" model to new waste removal and green waste elimination, few authors argue positively toward sustainability measures. Sustainability is positive to a CE, but CE implementation with sustainability measures are even better together instead of alone [99]. Nevertheless, CE implementation in the textile sector is not a simple task, as several challenges may exist during these initiatives, such as the fact that the textile sector depends on customer orders and raw material transportation costs. This indicates that there is a demand for favorable

laws for the textile sector from government authorizers. Textile companies should concentrate on the reduction of communication and raw material transportation costs and final products. Therefore, the present research contributes to the existing literature in a selective manner. It determined the comprehensive list of 12 key barriers to the implementation of CE in the textile sector. In addition, it classifies the most significant barriers to cause and effect groups. In addition, it shows the degree of interaction of key CE implementation barriers with each other.

This research aims to explore the DEMATEL approach applications in the analyzing of barriers to the adoption of CE actives in the textile sector. The DEMATEL model uses expert opinions to build a cause-and-effect diagram. Twelve barriers were identified based on a review of the literature and discussions with nine experts. The DEMATEL approach is used to explain the interrelationships with a causal relationship diagram and prominence (causal relationship graph). On this basis, the most significant barriers, the categories of the key barriers, and the group of causes and effects have all been identified. The twelve key barriers are divided into cause-and-effect groups. The aim of this study was to strengthen our understanding by determining the key barriers that could pose a critical adoption of CE in the textile sector. Among all the barriers, the most prominent barriers are B1, B3, and B4, as shown in Figure 3. However, from Figure 3, it is concluded that the barriers B1, B3, B4, B2, B6, and B7, out of which B1 has the highest impact value, cause these critical barriers. Similarly, B3 has the second-highest impact value. Therefore, it is suggested that B1 and B3 should be given due consideration. By comparison, "lack of support supply and demand network (B4)" is the least prominent or affected barrier based on the diagram of the causal interrelationships, as shown in Figure 3.

The results of this study provide significant practical implications and theoretical implications, which will help the managers and policymakers of textile companies to implement a CE. Prioritization and cause/effect-based analyses of key barriers will help textile companies' managers better understand and control the barriers to effectively implement CE policies for waste reduction and supporting the development of a sustainable business environment. The literature emphasizes the necessity to coordinate these measures to stimulate rather than impede organizational innovation [6]. Regarding the role of the organizational innovation process, this means that the supervisory framework needs to be regularly revised to ensure consistency even when policymakers are unaware of the innovation at the time of regulation. It is also important to ensure that the regulations do not impede innovation. For example, making alternative usages of waste is too complicated because of the high specificity of waste treatment. In addition, to textile companies' top management, there are also implications for policymakers and the wider public. This study has a number of implications, which are mentioned below.

Based on the DEMATEL result, it has been found that "consumers lack knowledge and awareness about reused/recycle (B1)" has strong interactions with another barrier. In addition, barriers such as "lack of successful business models and frameworks to implement CE (B3)" and "lack of support supply and demand network (B4)" are highly influencing barriers. This suggests that if the textile companies control the cost reduction of raw materials, the quality of the recycled martials will be needed to successfully incorporate CE practices in their supply chains [100]. This would need to be expressed by an ongoing production chain that encourages initiatives in CE practice and motivates employees to do likewise in order to achieve the overall goals of CE adoption in the textile sector [7]. The second most important factor is "lack of successful business models and frameworks to implement CE (B3)", which shows the significance of strong laws to reuse the production process of paramount importance. For instance, CE has been formally recognized by the Chinese government, as sustainable development plan, with effective implementation seen as a way to address the country's urgent problems of environmental destruction and resource scarcity [101]. Hence, the Taiwanese textile companies should develop strong policies for CE practices so that sustainable development can be achieved. Coming to theoretical implications, we have insufficient resources for a rapidly increasing global

population. However, we can better manage the scarcity of resources by managing waste. Textile companies should take strong measures to initiate and stick to reuse, recycle, and remanufacture policies in order to minimize waste. This CE approach aims to reduce resource consumption by reusing waste materials and reducing waste generation.

This study focuses on the key barriers in the Taiwan textile sector, which have been ignored in the present literature and validated by the literature review [87]. The existing research focuses either on the fundamental principles of CE or on other sectors. Theoretically, the current research contributes to the CE literature by identifying important CE implementation barriers, which are unique in the CE literature. However, there are certain limitations to this study. This research method is based on expert opinions, which could be biased. The initial-direct matrix obtained from the experts may, however, have been affected by the uncertainty of those relationships. Another limitation is that a combination of barriers to CE adoption in the textile sector may also be a constraint. In addition, it is focused on the Taiwan textile sector, which is subject to external generalization. In future research, scholars may expand on this work by examining a greater range of barriers in various sectors and regions, or focusing on using ANP to better understand the hierarchical interrelationships among CE implementation barriers. Scholars will develop on this work by prioritizing the barriers, as shown in Table 1, by using other relevant multi-criteria decision analysis approaches.

**Author Contributions:** Conceptualization, W.-K.C., V.N., M.-C.C. and C.-T.L.; Data curation, H.-C.H.; Formal analysis, W.-K.C. and H.-C.H.; Investigation, V.N. and M.-C.C.; Methodology, W.-K.C.; Project administration, C.-T.L.; Resources, M.-C.C.; Software, H.-C.H.; Validation, C.-T.L.; Writing—Review & editing, V.N. and C.-T.L. All authors have read and agreed to the published version of the manuscript.

**Funding:** This research was funded by Ministry of Science and Technology, Taiwan: 108-2221-E-212 -001 -MY2.

**Institutional Review Board Statement:** Not applicable.

**Informed Consent Statement:** Not applicable.

**Data Availability Statement:** We are providing the data.

**Conflicts of Interest:** The authors declare no conflict of interest.

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
