# Peer review of "Apply DEMATEL to Analyzing Key Barriers to Implementing the Circular Economy: An Application for the Textile Sector"

_applsci, doi:10.3390/app11083335_

Round 1

Reviewer 1 Report

The paper concerns the topical research problem of analyzing the barriers to implementing the circular economy concept in the industry. The paper is worth publishing, but the manuscript content needs to be improved. In particular, I suggest considering the following issues:

  1. The description of the research method is well structured, but its presentation needs improvements. It is not clear what the different figures used in the diagrams mean. Does the diamond symbolize questions? If so, they should have two exits. There is a lack of consistency in the number of stages given and the relationship between stages and steps. In Figure 1, there are three stages mentioned, but in the text, there is "stage 4" (line 356). In chapter 3, the steps are part of the stages, but in chapter 4, it is the other way round (lines 375). Figure 4 is not very readable. It does not show enough clearly presented relations between its elements. I recommend checking how this is presented by authors of other publications who used a similar research tool.
  2. The text contains unfortunate statements. It needs a detailed revision. For example:
    • Line 67. Does the “high use of greenhouse gases” is a major environmental effect of the textile industry? The use or emission is a problem?
    • Lines 254-255. Do companies or employees are experts?
    • The words 'export' and 'expert' are often used interchangeably.
  3. The language and style need to be deeply revised. I’m not a native speaker, but I found lots of unclear sentences.

There are also editorial errors in the text. The font style and size is not unified enough. The order in which the intervals are listed in table 2 is unclear.

Reviewer 2 Report

This study is highly relevant, especially considering the recent pandemic and the suggestion that our current consumption model has, at least in some way, contributed to the current state of global emergency. You have accurately described the textile industry as highly polluting and identified a number of the issues that must be addressed in order to implement a circular economy consumption model. You have also thoroughly explained the findings of a variety of authors as part of the extensive literature review. However, it is sometimes difficult to follow the discourse due to punctuation errors, repetition and other stylistic/grammar issues. The first c.a. 7 pages could be clearer and more concise. Additionally, although you correctly identified a number of factors and their degree of impact/influence on implementation of a circular economy model, the paper generally lacks analytical thought. Aside from the equations used to calculate impact, there is little explanation of the factors and how/why they determine a given effect. Adding such analysis would significantly improve the quality and usefulness of your work in order to assist decision makers in making the necessary changes to the current model.

Round 2

Reviewer 2 Report

Thank you for your modifications.